

# Magnetic locking in kerogen: impact on fluid transport

Benjamin Nicot1*, Jean-Pierre Korb2, Isabelle Jolivet1, Hervé Vezin3*, Didier Gourier4, Anne-Laure Rollet2

[1]TotalEnergies, Avenue Larribau, 64000 Pau, France
[2]Sorbonne-Université, CNRS, PHENIX, 75005 Paris, France
[3]Univ. Lille, CNRS, UMR8516 – LASIRe, F-59000, Lille, France
[4]Chimie ParisTech, PSL University, CNRS, Institut de Recherche de Chimie de Paris (IRCP), F-75005 Paris, France

*Correspondence to*: Benjamin Nicot (benjamin.nicot@totalenergies.com) and Hervé Vezin (herve.vezin@univ-lille.fr)

**Abstract.** How the transport of fluids in a confined and complex mixed organic/inorganic matrix can be far below the expected value from topological aspect? A good example of this situation is oil shales. Oil and gas shales are source

rocks in which organic matter has matured to form hydrocarbons. They exhibit a dual porous network formed by the intertwining of mineral and organic pores that leads to very low permeability. Still, the exact origin of this extremely low permeability remains somehow unclear. The present communication addresses this important question and provides novel insights on the mechanisms that strongly hinder fluid diffusion in such materials. By combining nuclear and electronic magnetic resonance techniques combined with SEM imaging, we evidence that magnetic locking occurs

in kerogen. This locking results from a magnetic coupling between vanadyl present in porphyrins and the organic matrix. We demonstrate that such coupling retards fluid diffusion and is reversible. This key dynamical feature explains the extremely low mobility of oil in shale rocks. This phenomenon may be a more general feature occurring in several systems where fluids are confined in a complex hierarchical matrix that embeds both organic and inorganic radicals resulting from ageing process.


## 1 Introduction

Millions or billions of years ago, the diagenesis of organic rich sediments led to the formation of rocks consisting of a mixture of minerals and solid carbonaceous matter called kerogen. These more or less hydrogenated carbonaceous

materials carry information about ancient life forms and their environments and are key for understanding the various attempts made during life evolution (Derenne, et al., 2008). Metabolism of most living systems, even the most primitive ones, are based on various metalloporphyrin complexes. Among the most important metal ions, $Mg^{2+}$ and $Fe^{2+}$ are involved in anoxygenic and oxygenic photosynthesis, and in anaerobic and aerobic respiration, respectively. During post-mortem degradation of the biological matter in the sediment, metal ions of porphyrins are substituted by

vanadyl ions $VO^{2+}$, giving very stable vanadyl porphyrines (VOP) (Breit, et al., 1991).



These organic-rich rocks are also of crucial economical interest as they are unconventional oil and gas source rocks. Furthermore it requires the use of hydraulic fracturing to compensate the extremely low mobility of hydrocarbons yielding adverse environmental consequences (Spellman, 2013).

An accurate petrophysical evaluation of these rocks (volume of resources in place: porosity, hydrocarbon saturation, permeability and fracability) has appeared more challenging in tight organic shales than in conventional reservoirs (Sondergeld, et al., 2010; Le Bihan, et al., 2014). In particular, the evaluation of water/hydrocarbon saturations (the fraction of porosity filled by water or hydrocarbon) using classical "Dean Stark" and "Retort" techniques have proved to be inaccurate (Handwerger, et al., 2011; Handwerger, et al., 2012; Simpson, et al., 2015). Recently, low field
Nuclear Magnetic Resonance (NMR) has solved this issue using two dimensional T1-T2 NMR correlation maps (Nicot, et al., 2015). Very high T1/T2 ratios have been found experimentally both on shale samples (Nicot, et al., 2015) and kerogen isolates (Singer, et al., 2016). The origin of such very high T1/T2 ratio and its dependence on Larmor frequency has been explained using field cycling NMR relaxometry (Korb, et al., 2014). Now remains the issue of hydrocarbon transport and its relationship with the multiscale structure of kerogen.


In the present communication, we aim at addressing three fundamental questions linked to these non-conventional microporous materials. (i) What is the main physical origin for the extremely low permeability of these materials? (ii) How to localize fluids either in mineral or organic porosities? (iii) Finally, how to characterize their individual dynamics?


To address these issues, we combined the use of original multiscale and multidimensional nuclear magnetic relaxation and advanced pulsed Electron Magnetic Resonance (EMR) with imaging techniques. We first describe the multiscale intertwined structure of the shale samples under study. We then focus on the multiscale structure of kerogen, and we observe kerogen swelling in the presence of oil, through a modification of paramagnetic nano-structures. We observe
a magnetic locking phenomenon appearing within the solid kerogen structure as a result of a coupling between vanadyl ions and a carbon radical. This is a key feature that could explain the extremely low mobility of hydrocarbons in shale rocks. Finally, we reveal the nature of both fluids' dynamics locked in this complex porous medium.

## 2. Methods

### 2.1. Samples:

The samples studied here are shales coming from the Late Jurassic Vaca Muerta Formation, located in the Neuquén Basin in northern Patagonia, Argentina. This formation hosts major deposits of shale oil and shale gas. Three samples (diameter 10 mm, length 16 mm) were studied and yielded similar results. Samples were studied in different states: the "as received" state, in which the shale contains the native fluids; the "dry" state in which the rock does not contain
any fluid. The kerogen was then isolated by HCl/HF acid demineralization process. Experiments presented here



include experiments on "dry" kerogen isolates and on "dodecane-impregnated" kerogen isolates (after submitting the kerogen isolates to dodecane).

## 22. Electron Microscopy:

To perform elementary micro-analyses and nano-structural observations by SEM (Scanning Electron Microscopy) coupled with EDS (Energy Dispersive X-ray Spectrometry), the samples were polished mechanically down to ¼ micron with diamond suspensions and then ion milled with the Fischione 1060 at 5KV and with an argon gun at tilt angles of 5° and 2°. To ensure a good conductivity on the sample surface and maintain a good image quality for SEM observations, the plugs were coated with platinum.

To assess the rock heterogeneity a quantitative mineralogical map was acquired on a FEI Quanta 650 electron microscope equipped with 2 Energy Dispersive X-Ray spectrometers (EDS Bruker X-Flash) and combined with the mineral identification software package Maps-Nanomin. The interpretation method used in the Nanomin software was previously calibrated with quantitative data based on X-Ray diffraction and X-ray Fluorescence (Fialips, et al., 2018). Additional SEM observations at different length scales were then performed locally to qualitatively identify the

different types of porosity in the organic matter and in-between the minerals following the classification proposed by Louks (Loucks, et al., 2012).

To quantify the porosity by image processing from the nanometric scale up to the micrometric scale (typically in an equivalent diameter range from 10 nm to few micrometers), large area imaging was performed on a Zeiss Crossbeam 540 electron microscope. Backscattered electron images were acquired at low acceleration voltage (5KV) and a current

beam of 100-200pA to cover a representative area of 480 µm x 240 µm at a resolution of 5 nm (pixel size). Segmentation of organic matter and pores was then performed on filtered images using the Ilastik software. The classification of the different types of porosity (mineral vs organic matter hosted pores), performed using the software Visilog, was then carried out using current image processing based on mathematical morphology, where the pixels describing the pores neighboring the organic matter are detected after a dilation operation. The mineral hosted pores

were then deduced by subtraction. From the image, the surface of each pore is extracted whatever its shape, a diameter is calculated for an equivalent disk. The pore size distributions for each pore types and the derived fractal dimensions were calculated from this image classification (cf. fig SEM4).

Further observations at higher resolution were also performed on the Zeiss Crossbeam 540 electron microscope on ultra-thin sections of roughly 100nm of thickness, using the SEM-STEM (transmitted and scattered electrons detector

allowing observations in transmission in a scanning electron microscope with an optimized resolution under the nanometer). The main interest of this technique is to complete the SEM imaging to go further in the qualitative description at higher magnification of the porous network in the organic matter (Figure 3).



### 2.3. NMR relaxation measurements:

2D $T_1$-$T_2$ maps were acquired at 2.5 MHz and 23 MHz on Oxford Instrument spectrometers with inter-echo time TE = 200 μs and inversion recovery varying from 70 μs to 1s in 200 values. The results were processed using a 2D inverse Laplace transform (Venkataramanan, 2002). The temperature of the samples was 21±1 °C. Measurements were performed at 2.5 MHz on cylindrical samples of 30 mm diameter and 50 mm height, while measurements at 23 MHz measurements were performed on samples of 10 mm in diameter and 15 mm height.


### 2.4. Fast Field Cycling NMR Relaxometry:

    Multi-frequency NMR relaxation dispersion of longitudinal relaxation rate (NMRD) was performed on a fast-field cycling spectrometer from *Stelar s.r.l., Mede, Italy*. The main interest of this NMR technique is to explore a large magnetic field range allowing sensing a large range of fluctuations to which the nuclear spin relaxation is sensitive in

confinement. The measurements were performed on samples 9 mm in diameter and 15 mm in height. At each magnetic field associated to a $^1$H Larmor frequency, varying from 10 kHz to 35 MHz, a measurement of the longitudinal relaxation time $T_1$ is performed and processed using an in-house 1D inverse Laplace transform leading to a bimodal $T_1$-distribution allowing separating the oil and brine respective contributions.

**2.5. Electron Magnetic Resonance:**

    The measurements were performed on samples of 8 mm in diameter and 15 mm in height for continuous wave (CW) experiments and a piece of 3x3mm dimension for pulsed EMR experiments. Continuous wave EMR spectra (CW-EMR) were recorded at the X-band (≈ 9.4 GHz) at room temperature using a Bruker ELEXSYS E500 spectrometer equipped with a 4122SHQE/011 resonator. Pulsed-EMR experiments were carried out at 5K with a Bruker ELEXSYS

E580 X-band spectrometer equipped with a Bruker cryostat "cryofree" system.

    Hyperfine Sublevel Correlation pulse sequence (HYSCORE) (Hofer, 1994) was used to reveal hyperfine interactions of the electron spins of carbonaceous matter with $^{13}$C (I=1/2; 1.1 % abundance), $^1$H (I=1/2; 100% abundance), and $^{29}$Si (I=1/2; 4% abundance) nuclei. In this technique, a spin echo is generated by the pulse sequence $\pi/2$-$\tau$-$\pi$/2-$t_1$-$\pi$-$t_2$-$\pi$/2-$\tau$-*echo*. The angles $\pi/2$ and $\pi$ represent the flip angles of the electron magnetization. Its intensity is measured

by varying the times $t_1$ and $t_2$ at constant time $\tau$ in a stepwise manner. The lengths of the $\pi/2$ and $\pi$ pulses were fixed at 16 ns and 32 ns, respectively. 256x256 data points were collected for both $t_1$ and $t_2$ at increments of 20 ns. $\tau$ value was set at 136 ns for all the sample. The unmodulated part of the echo was removed by using second-order polynomial background subtraction. The magnitude spectrum was obtained after 2D Fourier transformation of the spectra by using a Hamming apodization function.


    For the PELDOR experiments, a 4-pulse sequence with Gaussian, non-selective observer and pump pulses of 8 or 16 ns length with 280 MHz frequency separation was used. An eight-step phase cycling was performed together with





0–π phase cycling to remove unwanted effects of running echoes from the DEER trace. The evaluation of the DEER data was performed using DeerAnalysis2018 (Jeschke, et al., 2006). The background of the primary DEER traces was

corrected using exponential functions with homogeneous dimensions. A model-free Tikhonov regularization was used to extract distance distributions from the background corrected form factors (Jeschke, et al., 2002).

## 3. Results

### 3.1 Multiscale intertwined structure of organic and inorganic porosities

In general, shales are fine-grained and laminated sedimentary rocks consisting in a mixture of several minerals (clays, quartz, calcite, feldspars, pyrite, etc.) and solid organic matter (kerogen) and pores. These pores can be located in the mineral phase and in the organic matter and can contain either hydrocarbons or water. Understanding fluid transport in such complex structures requires the knowledge of their hierarchical organization from nanometer to micrometer scales, and of the intertwined nature of the porous network.


Scanning electron microscopy was performed on the samples in order to reveal their microstructure. Making successive zooms (Figure 1) allows studying the structure of the sample at various scales, taking into account samples heterogeneity. The rock is composed of pure minerals grains, and a nanotextured mud formed by a mixture of clays, micro quartz, calcite debris and organic matter.


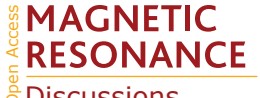

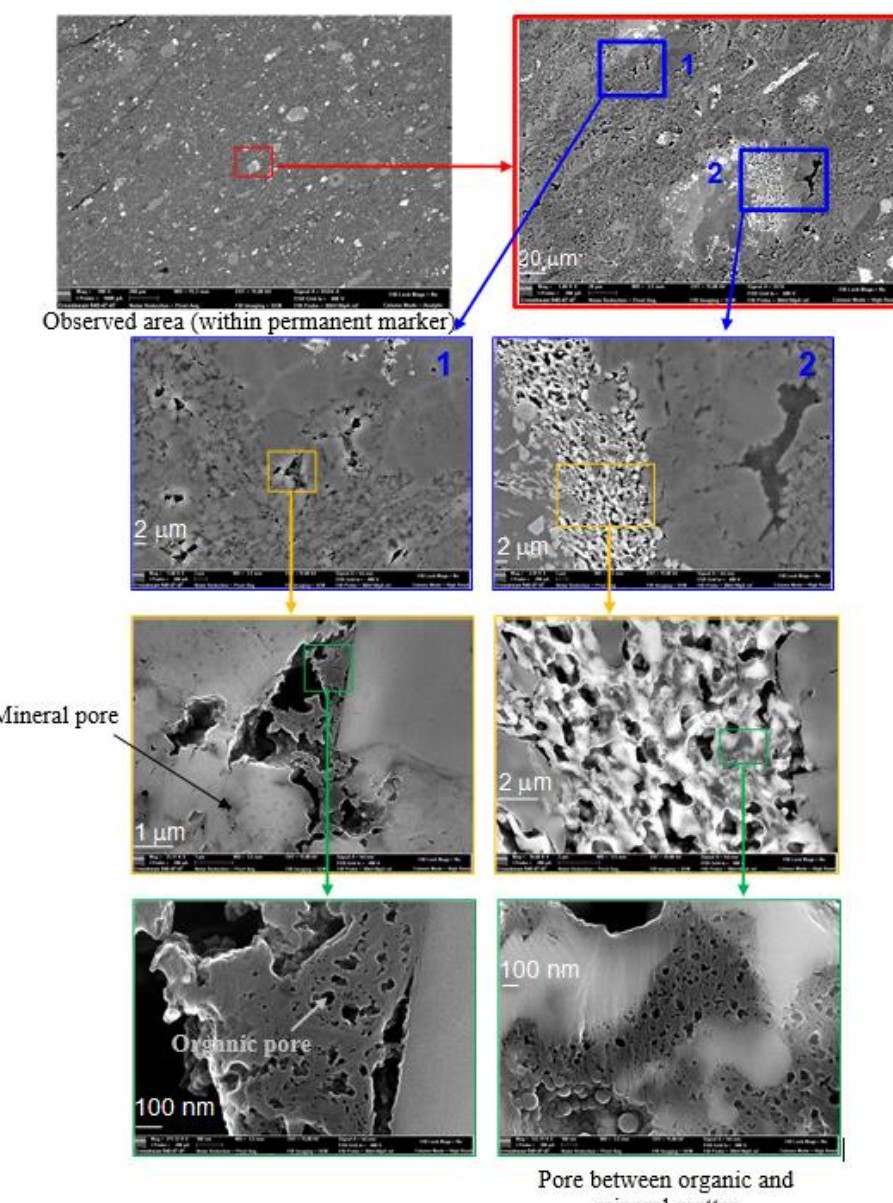

*Figure 1: Visualization of the microstructure and the arrangement of mineral grains versus organic matter using SEM imaging at different length scales from 20 μm to 100 nm (from top to bottom). The successive zooms focus on the meso and macro porosity in the organic matter.*


Several different types of porosity can be identified at the micrometric scale. The two main types observed in organic matter are the following (all porosity types are described in supplementary material): a sponge like organic matter (left panel in Figure 2) and a lamellar organic porosity, intricated in the clay structure (right panel in Figure 2).



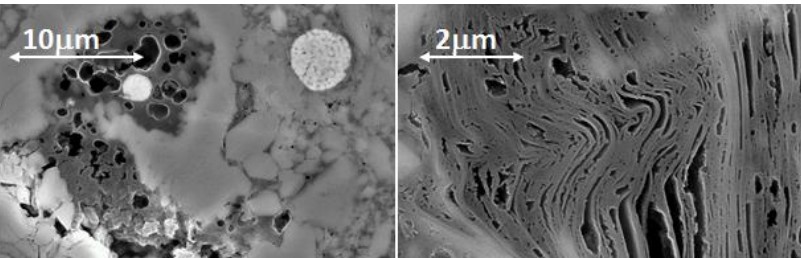


*Figure 2 SEM images of shale samples at different resolution. On the left a "sponge like" kerogen porosity and a framboidal pyrite, on the right a lamellar clay structure filled by porous kerogen.*

Interconnected pores networks appear to be present rather in the nanostructured mud areas. Further investigations

were carried out by SEM-STEM in these mud areas to study the nanostructure of the porous network, as shown in Figure 3. Typically, we evidence a tight network of mesopores and macro-pores that can be connected by nano pore throats of few nanometers away from each other.

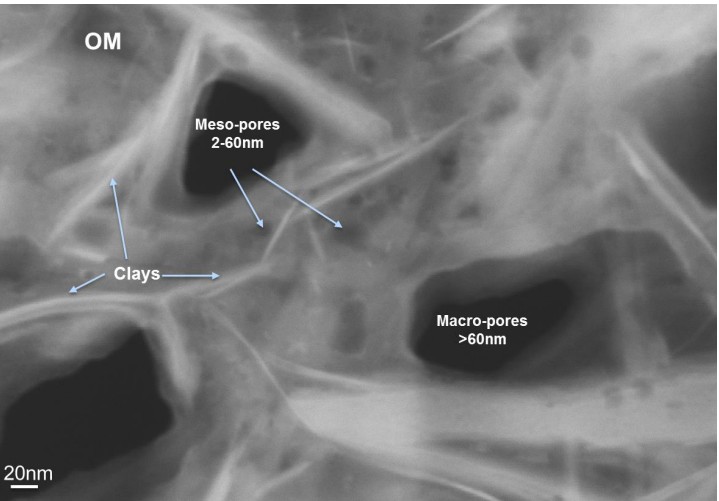


*Figure 3 SEM-STEM imaging performed on a mud area enriched in organic matter (OM) and clays. Visualization of macro and meso-porosity of few nanometers of equivalent diameters, which are either connected or distant of few nanometers*

Advanced image processing of zoomed SEM images allows thresholding the image to distinguish organic matter,

organic porosity (which represents 75 to 85% of the porosity visible on SEM) and mineral porosity. The relation between the number of pores $N(R)$ and pore size $R$ is a power law $N(R) \propto R^{-Df}$, where $D_f$ is the surface fractal dimension. A fractal dimension of about 2.3 was found for the three samples tested, in agreement with Curtis et al. (Curtis, et al., 2010). This provides a quantitative measurement of the hierarchical character of the porous



microstructure already shown in Figure 1. Therefore, electronic microscopy reveals the hierarchical structure of

kerogen, with porosity appearing as a sponge-like fractal, either in patches or filling the lamellar structure of clay

minerals.

Since clays and kerogen are well known for containing paramagnetic species, the structure of such complex materials

can also be investigated by quantitative EMR. Continuous wave (CW) EMR spectrum (Figure 4a) unambiguously

reveals the presence of paramagnetic $Mn^{2+}$ ions (six hyperfine lines) and an organic radical labelled $C^o$ (intense single

line centered at g=2). By integrating these calibrated spectra, we found $\approx 4.50 \ 10^{19}$ $Mn^{2+}$ and $\approx 1.20 \ 10^{17}$ carbon radicals

($C°$) per gram of rock. Moreover the continuous wave spectrum of extracted kerogen (shown in supplementary

material) shows only the single line of carbon radicals around g=2. This proves that the carbon radicals are located in

kerogen while manganese impurities belong to the mineral phase.


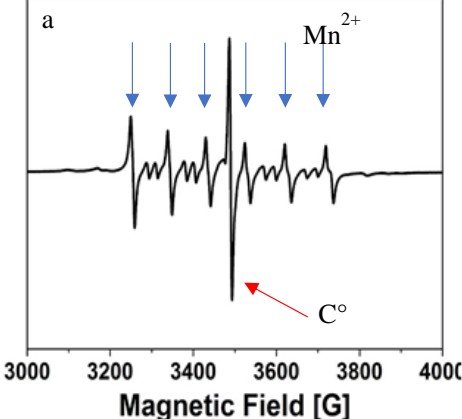

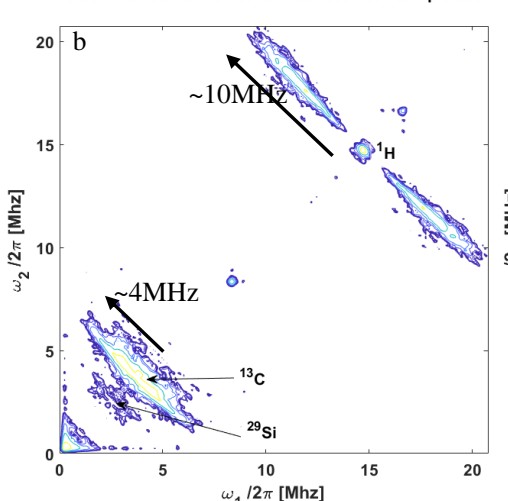

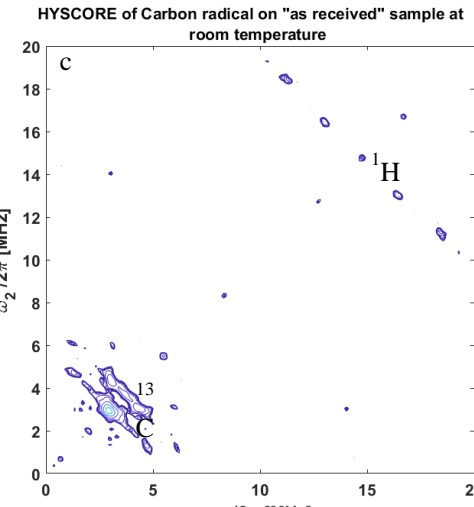

*Figure 4 EMR experiments performed on the rock sample in the "as received" state. a- Continuous wave EMR spectrum performed at room temperature; EPR lines are indicated by blue arrows for Mn2+ and a red arrow for the carbon radical C⁰. b-: 2DEMR HYSCORE of carbon centered radical C⁰recorded at 5K; The inset shows the echo detected EMR signal. c-: 2DEMR HYSCORE of C⁰ recorded at room temperature.*

2D HYSCORE (HYperfine Sublevel CORrElation Spectroscopy) (Hofer, 1994) experiments performed on "as received" shale samples (Figure 4b) reveal the local nuclear environment of $C^0$ radicals ($^{13}C$ at 3.7MHz and $^1H$ at 14.5MHz). The nature of the neighboring coupled nuclei (their Larmor frequency) is measured along the first diagonal, and the strength of their couplings is measured along the anti-diagonal (4 and 10 MHz for $^{13}C$ and $^1H$, respectively). Such values are typical of what can be measured in primitive organic matter (Gourier, et al., 2008).

As will be demonstrated below, the observed $^1H$ signal mainly arises from the interaction between the kerogen radical $C^o$ and hydrogen atoms of trapped oil. The fact that such interaction is visible in the 2D spectrum at 5K requires a quasi-static environment of the protons around the radical. Moreover, the $^1H$ pattern intensity decreases when recorded at room temperature but does not fully vanish (Figure 4c). This points to a very low mobility of the oil protons as a hyperfine coupling of ≈10 MHz comparable with that found at 5K can still be observed at room temperature.

Once the paramagnetic species have been identified and quantified, we address the question of the uniformity of their spatial distribution. EMR spatial and spectral/spatial ($C^\circ$ or $Mn^{2+}$) imaging (shown in supplementary material) reveals a homogeneous distribution of paramagnetic species, and a nearly constant $C^\circ/Mn$ ratio at a spatial resolution of 1μm. This proves that despite the extreme heterogeneity of the sample, the sources of NMR relaxation (paramagnetic impurities) are homogeneously distributed within the sample.

### 3.2 Interactions between oil and kerogen

Having identifed the two types of paramagnetic species ($Mn^{2+}$ in minerals and $C^o$ radicals in kerogen), we used them to probe the relations between the three components of the shale: minerals, kerogen and oil. Considering the intertwined structure of clays and kerogen evidenced by electronic microscopy, we performed EMR experiments to probe the $Mn^{2+}$ - $C^o$ interactions between clays and kerogen.

We performed Pulsed ELectron Double Resonance (PELDOR) experiments   (Jeschke, et al., 2006), which allows calculating distances between paramagnetic centers by refocusing the dipolar interaction between two paramagnetic centers, namely $Mn^{2+}$ in the mineral phase and $C^o$ in solid kerogen. Figure 5 shows the distribution of $Mn^{2+}$-$C^o$ distances obtained in dry shale and in the shale impregnated with dodecane. Three sets of distance-distributions centered on 3.5, 4.5 and 6.0 nm are measured in the dry shale, while adding dodecane to the shale results in a significant shortening of these distances (broad distribution around 2.5 nm, 3.7 and 4.8 nm). The striking point is that the structure is conserved upon swelling. Indeed, these distances (with or without dodecane) are consistent and provide an estimate of the mean Mn-Mn distance of about 2nm. It is worth pointing out that a similar result can be obtained by estimating the Mn-Mn distance from the overall Mn concentration ($\eta_S$=4.5 $10^{19}$ Mn/g of rock), the rock density ($\rho_S$≈2.6 g/cm³) assuming a uniform spatial distribution.





This PELDOR experiment reveals that dodecane penetrates the nanostructure of solid kerogen inducing swelling,
reinforcing the dipolar interaction between a C° radical in the kerogen and $Mn^{2+}$ in the mineral phase. Although
swelling has been previously observed at the macroscopic scale (Ertas, et al., 2006), this is the first time that swelling
is experimentally observed and quantified at the nanometric scale. Moreover, we verified that this phenomenon is
reversible: after drying out all the dodecane, the 2D HYSCORE spectrum is the same as the spectrum obtained before
dodecane impregnation.


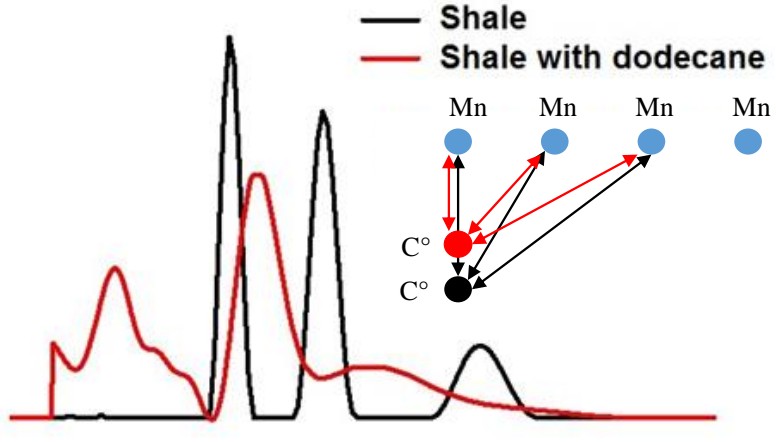

*Figure 5 Distribution of carbon (C°)-Manganese distances in the the dry shale sample (black) and after spontaneous imbibition of dodecane (red), measured at 40 K using four pulses PELDOR sequence. All the distances measured are consistent with the schematic nanostructure of paramagnetic species proposed in the inset.*


### 3.3. Magnetic locking of oil inside solid kerogen

In order to probe potential interactions between oil and kerogen, experiments have been performed on dry kerogen,
and on kerogen impregnated by dodecane.

Figure 6a displays the echo field sweep EMR spectra of extracted kerogen, exhibiting a typical single line of 5 G
linewidth characteristic of C° radical. Additional information can be extracted from the line shape of this signal. The





ratio of Gaussian/Lorentzian character indicates that the shale has approximately 130 M years, a value close to those found using classical datation estimates, i.e. from 140 to 150 M years (Tomassini, et al., 2016). In presence of dodecane impregnation, one observes a drastic increase of the $C^o$ signal and the appearance of a new signal of vanadyl $VO^{2+}$

ions (Figure 6b)

The variation of intensity of the $C^o$ signal as a function of temperature in a range from 40 K to 5 K (Figure 6c) unambiguously characterizes a weak antiferromagnetic behaviour of the pure extracted kerogen. This indicates the presence of pairs of radicals with S=1/2 interacting by a weak exchange interaction, giving two states with total spin S=0 and 1. The state S=0 is at lower energy with a J value of -0.2 $cm^{-1}$. On the other hand, after kerogen swelling with

oil, the $C^o$ radical displays a Curie's type paramagnetism), *i.e* its intensity increases as 1/T with decreasing temperature. The simultaneous increase of $C^o$ and the appearance of $VO^{2+}$ signal can be explained by a structural ordering (antiferromagnetic interaction) between $VO^{2+}$ and $C^o$ of the kerogen, both having S=1/2. The presence of oil induces kerogen swelling that breaks the antiferromagnetic interaction between $VO^{2+}$ and $C^o$ radicals, resulting in a pure curie's behavior of $C^o$ radicals. Analysis of the $VO^{2+}$ signal by 2D HYSCORE reveals a nitrogen pattern typical

of vanadyl porphyrins, an ubiquitous paramagnetic complex of bitumen and oil (Gourier, et al., 2010; Ben Tayeb, et al., 2015; Ben Tayeb, et al., 2017).

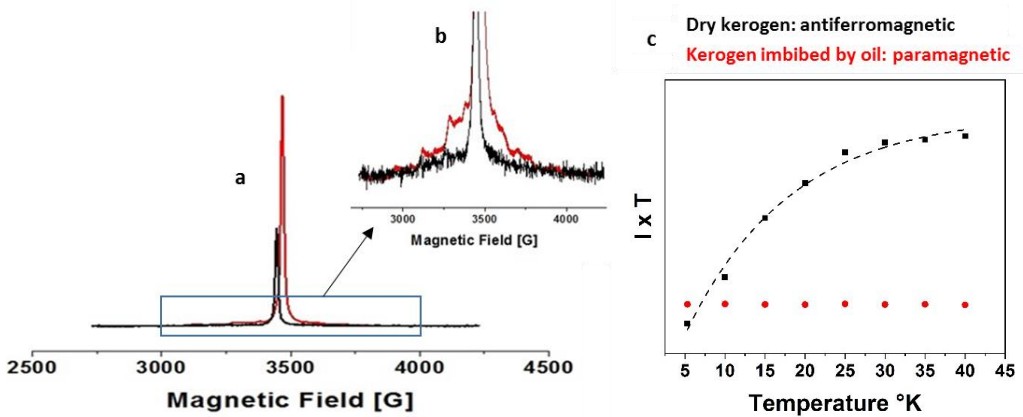

*Figure 6 a-Pulses echo field sweep experiments of dry extracted kerogen (black) and kerogen impregnated with dodecane (red)*
*recorded at 5K. b- Zoom on the vanadyl porphyrin moeity. c- I x T plot of $C^o$ intensity I versus temperature T between 5 K and 40 K for the extracted kerogen (in black) and for the kerogenimpregnated with dodecane (in red).*


In order to bring to light the interaction between $C^o$ radicals of kerogen and nuclei of dodecane, we investigated their magnetic couplings using 2D-HYSCORE experiments in the dry state of kerogen compared with the dodecane impregnated state.



For dry extracted kerogen (Figure 7a), only a weak signal with small couplings with $^{13}$C (at 3.7 MHz) and $^{29}$Si (at 2.9
MHz) is observed. The absence of proton coupling indicates the H/C ratio is low and that the closest protons are at
least 5Å away from the carbon radical.  This indicates a high level of maturity of the organic matter (Gourier, et al.,
2010) (Ben Tayeb, et al., 2015) (Ben Tayeb, et al., 2017). The presence of silicon can be explained as a mineral residue
of the acid attack (demineralization) necessary for extracting kerogen.

After oil impregnation (Figure 7b), the spectrum exhibits typical proton and carbon patterns, quite similar to the one
obtained on the "as received" shale (Figure 4). This implies that the protons and carbons nuclei interacting with C°
belong to the dodecane molecules. Therefore this experiment evidences the very close proximity of dodecane molecule
to the carbon radicals in the heart of the kerogen. This proves that oil molecules are located within organic pores in
the solid kerogen. The similarity of the 2D-HYSCORE spectra of dodecane impregnated kerogen (Fig.7b) and of the
pristine shale (Fig. 4a) demonstrates that in the later, the oil molecules are located in the pores of the kerogen
component of the shale. This interpretation is also confirmed by the absence of carbon or proton features on a
HYSCORE experiment performed on Mn$^{2+}$ transition (Supplementary Material).

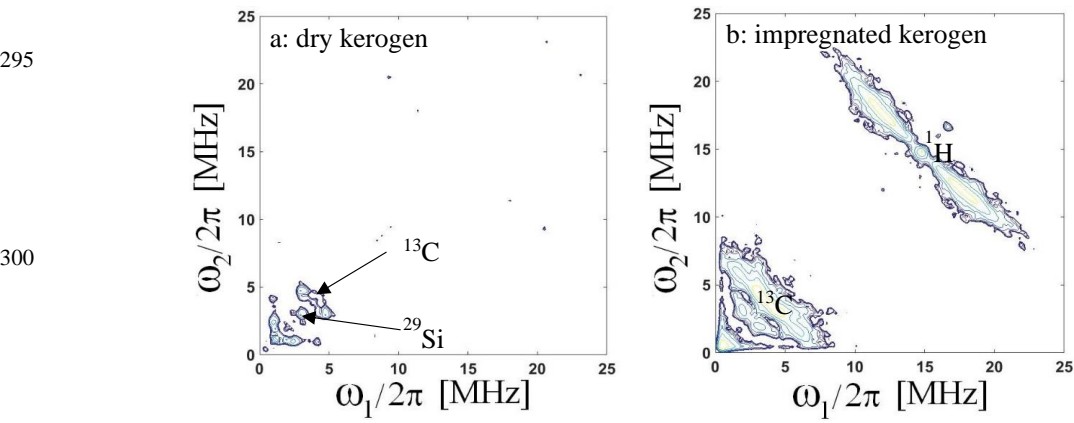

*Figure 7: a- 2D HYSCORE of carbon centered radicals of extracted Kerogen,  b- extracted kerogen impregnated with dodecane model oil
(right). All spectra are recorded at 5K*

After oil removal by evaporation under vacuum, the proton and carbon patterns disappear (supplementary material).
The structure of the kerogen network is therefore reversible.

All these EMR experiments provide a comprehensive image of all the interactions that are present in the organic part
of a shale:

- Two paramagnetic centers are identified in kerogen: a carbon radical C° and a Vanadyl porphyrin;





- In a dry kerogen, the C° radical and the Vanadyl porphyrin are coupled in an antiferromagnetic state;
- Oil impregnation breaks this antiferromagnetic coupling;
- Oil molecules are in very close vicinity of the carbon radical, and therefore located in the pores of the kerogen.

All these features imply a magnetic locking of oil molecules in between the carbon radical and the Vanadyl porphyrin.


The identification of the type of paramagnetic centers and their quantity is also a key for the interpretation of the fluid dynamics investigated by nuclear magnetic relaxation dispersion (NMRD) experiments described in the following section.


### 3.4. Evidence of hindered fluid dynamics

In order to assess the physical impact of this magnetic locking, it is important to probe accurately the dynamics of the liquids (oil and water) *in situ* and non-invasively. The NMR fast field cycling technique is perfectly suitable for this purpose (Kimmich, 1997). It explores a large range of Larmor frequency $\omega_0/2\pi$ and correspondingly senses longer

correlation times of the dipolar fluctuations that are induced by liquid dynamics at the origin of the nuclear magnetic relaxation dispersion (NMRD) of the longitudinal spin-relaxation $R_1(\omega_0)$ (Figure 8a). Analyzing the NMRD profiles thus allows separating directly the surface dynamics of water and oil in micropores. Figure 8a displays the very different observed NMRD profiles associated to these two fluids embedded in an as received shale. It represents the longitudinal relaxation rate $R_1 = 1/T_1$ versus the Larmor frequency. In a previous paper, we succeeded in identifying

the NMRD profiles of these two fluids that cross each other around 1 MHz (Korb, et al., 2014). The brine (blue continuous line) exhibits a quasi-logarithmic frequency dependence whereas oil (red continuous line) follows an inverse square-root behavior with a leveling-off at low frequency (details in the Supplementary Material).





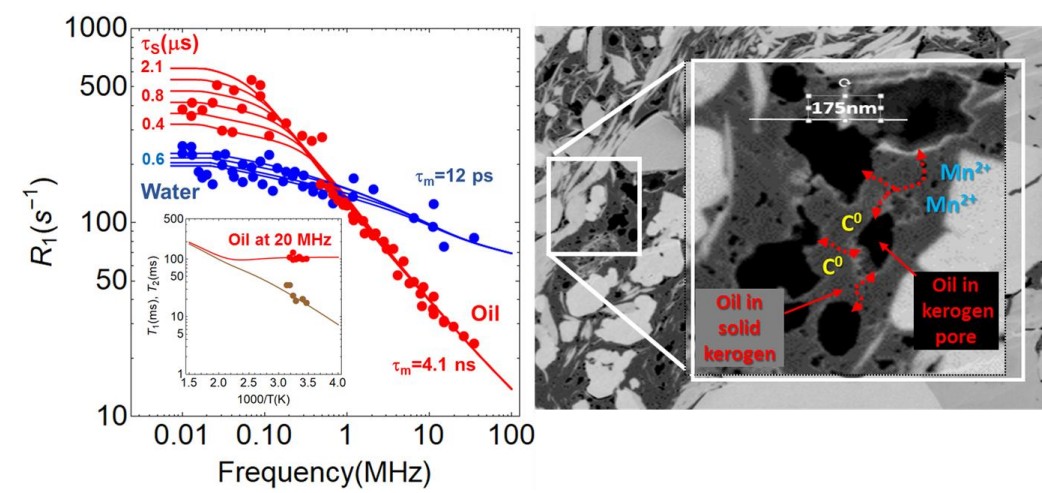


*Figure 8: a - NMRD profiles of liquids obtained on an "as received" shale (oil in red, water in blue), representing the variation of the longitudinal relaxation rate $1/T_1$ versus the frequency. The inset shows the temperature dependence of relaxation times $T_1$ (red) and $T_2$ (brown) at 20MHz for oil obtained on another "as received" shale. The continuous lines are the best fits obtained with the model described in Supplementary material, with $\tau_m$ and $\tau_s$ the translational and surface residence times, respectively. b-*

*Representation of the diffusion-relaxation modeling of oil in the connected pores of kerogen.*

In order to interpret these profiles unambiguously it is crucial to use a relevant theoretical nuclear-spin-relaxation model. Here, we just outline the main two features of the used relaxation model described in the supplementary material document. First, we have observed a mono-exponential decay for each fluid, proving a biphasic fast exchange

between proton populations at the pore surfaces and in the bulk. Second, the translational diffusion of both fluids modulates the heteronuclear dipole-dipole interaction between the mobile proton species (water or oil) and the different paramagnetic species fixed at pore surfaces.

The NMRD profile for brine reveals an NMR relaxation induced by a two-dimensional diffusion in the vicinity of

$Mn^{2+}$ (S=5/2) at surfaces of lamellar clay mineral (see Figure 8b) (Korb, et al., 2018). This yields an estimate of the water translational diffusion coefficient at the mineral clay-like surface $D_{surf}$=1.9 $10^{-9}$ m²/s for a specific surface area of clay $S_p$=47 m²/g. This local diffusion coefficient is similar to the one of bulk brine and shows that at the local level, the dynamics of water molecules is not hindered (Mills, et al., 1989).

On the other hand, the particular frequency dependence observed for oil strongly suggests a relaxation process induced by a highly confined translational diffusion. The model used for interpreting the oil NMRD profiles relies on a quasi-1D translational diffusion of oil at the vicinity of paramagnetic sources of relaxation. We used the nature and concentration of paramagnetic centers ($C^0$ and $VO^{2+}$), as well as the total spin states S=1 for the $C^0$ - $VO^{2+}$ pairs determined by EMR.




With such inputs, the model is able to fit quite well experimental results, giving information on the dynamical parameters: (i) a specific surface area $S_p$=233 m$^2$/g, (ii) a translational diffusion correlation time $\tau_m$=4.1 ns, (iii) a relevant kerogen pore size (R= 0.3 nm) at the maximum of the pore size distribution N(R), and (iv) a very slow translational diffusion coefficient $D_{surf}$=2.6 $10^{-7}$ cm$^2$/s of oil.


To assess the reliability of the NMRD data analysis, we made two supplementary verifications of the proposed model. First, the temperature dependencies of the longitudinal $T_1$ and transverse $T_2$ relaxation times of oil at 20 MHz for a second sample are displayed in the inset of Figure 8. As detailed in the Supplementary Material, the asymptotic theoretical temperature dependencies of these relaxation times behave as: $T_1 \propto \sqrt{(\omega_I/\tau_m)}$ =C$^{te}$ and $T_2(T) \propto 1/\sqrt{(\tau_m\tau_s(T))}$

as it observed on the experimental data. Here $\tau_m$ and $\tau_s$ are the translational and surface residence times, respectively. Second, 2D spin correlation maps $T_1$-$T_2$ for oil and brine embedded in shale oils rocks at 2.5 and 23 MHz are displayed in supplementary material. On the same figure the theoretical evolutions calculated at these Larmor frequencies for $\tau_m$=4.1 ns with 0.4 $\mu$s < $\tau_s$ < 2.1 $\mu$s for oil and $\tau_m$=12 ps with 0.4 $\mu$s <$\tau_s$ < 0.6 $\mu$s for brine are superimposed and show a quite satisfactory agreement. This 2D NMR measurements are universally used allowing a fluid-typing

downhole in petroleum wells.

Moreover, low frequency data in Figure 8a are rather dispersed, which can be due to different values of the activation energy associated to the time of residence $\tau_s$ ranging between 0.4 and 2.1 $\mu$s. This observation is consistent with the simulation of Lee et al. (Lee, et al., 2016) who consider a wide distribution of residence times for oil in kerogen

nanostructure that inhibits the activated desorption of this fluid.

Finally, a very fast penetration of oil within the kerogen has been observed (Nicot, et al., 2015). This proves that the different kerogen patches are well connected to each other (Figure 8b). However, due to the magnetic locking occurring in kerogen, oil is transferring by diffusion very slowly between large organic pores through solid kerogen.

This key dynamical feature thus answers clearly the fundamental questions concerning the origin of the extremely low permeability of shale rocks.

### 4. Discussion / Conclusion:

The main results can be summarized as follows. The complex hierarchical structure of shale porous network has been revealed by electronic microscopy and shows the dominant contribution of nanopores. The diffusive nature of both water and oil motion in shales has been proved by NMRD experiments, showing the extreme confinement of oil in kerogen. Kerogen swelling at the nanoscale has been evidenced by EMR spectroscopy performed on kerogen isolates. A magnetic locking has been discovered, whereby hydrocarbon molecules are locked between carbon radicals C$^0$ and

vanadyl porphyrin ions. The reversibility of this magnetic locking has been evidenced by the appearance and



disappearance of a vanadyl signal when dodecane is imbibed in the rock or dried out. These findings are summarized in Figure 9.

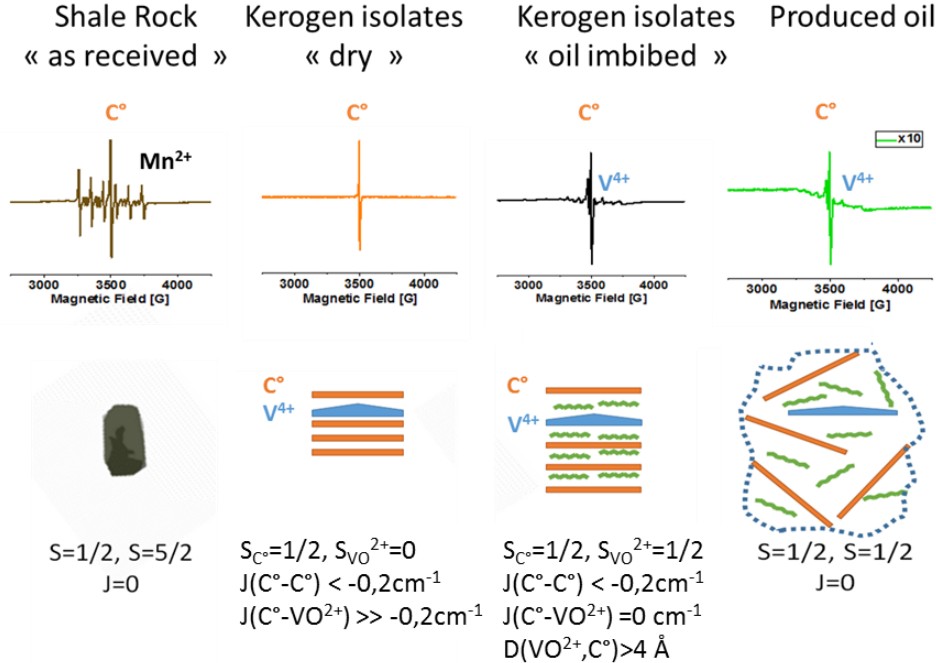


*Figure 9: Summary of the different situations encountered in the studied samples, with CW EMR spectra at room temperature (top), scheme of magnetic interactions and related structures (middle), and spin states values and exchange interaction parameters (bottom). Estimated distances between vanadyl and carbon radical ($d_{V4^{2+}-C}$) are also given. In the "as received" shale sample, only two paramagnetic species can be observed (carbon radical $C^0$ and manganese $Mn^{2+}$); in the dry kerogen isolate, only the*


*carbon radical signal is observed, whereas in the dodecane-imbibed kerogen isolates, the appearance of a vanadyl signal is observed. These two results demonstrate that the $C^0$ and $VO^{2+}$ are antiferromagnetically coupled. This interaction is broken by oil intercalation. The keys of the magnetic locking arising from the vanadyl porphyrin complexes which are coupled with the organic matter. Finally, the analysis of the crude oil produced from this shale shows the presence of small amounts of both organic radicals and vanadyl.*

However, vanadyl ions are observed only on the kerogen isolates, never on a shale even if the swelling is observed (Figure 5). Therefore, it seems relevant to think that kerogen swelling is spatially limited in the rock due to the presence of minerals, preventing the inhibition of magnetic locking.

Moreover, these results reflect the high degree of structuration of kerogen, with alternating stacks of kerogen (containing $C^0$ radicals) and vanadyl porphyrins as sketched schematically in Figure 9. The magnetic interaction

between these stacks could play the role of a real magnetic locking, prohibiting the collective diffusion of oil, and therefore preventing long distance fluid transport. This key dynamical feature explains the extremely low mobility of oil in shale rocks.



Finally, these results and hypotheses are strongly supported by the fact that the EMR spectrum of the bulk extracted oil reveals the presence of vanadyl porphyrin indicating that the fracking method extraction is sufficiently powerful to break this magnetic interaction and release a fraction of vanadyl content in the extracted oil.

The reversible magnetic locking revealed here by joint NMR and EMR techniques in shales might be a more general phenomenon occurring not only in geological system where organic matter is degraded in confined rocks but also in various systems where ageing processes result in the formation of organic and inorganic radicals. Therefore, the proposed approach could open new areas in various fields where the ageing of organic matter is of key interest such as food industry (ageing over weeks), archeological objects (ageing over hundreds or thousands of years), nuclear waste storage (ageing over millions of years) …

**Author contributions:**

H.V. performed the EMR experiments and interpretation. J.P.K. and A.L.R. performed the NMRD experiments and theoretical interpretation. I.J. performed the electronic microscopy and interpretation. B.N performed the 2D NMR experiments. All the authors contributed to the discussion and article redaction.

**Acknowledgements**:

We acknowledge the CNRS infrastructure RENARD (FR 3443) for EMR facilities.

We acknowledge the Institute of Materials of Paris (IMPC) for the access of the NMR relaxometers (RELAXOME facility). The NMR relaxometers were funded by Sorbonne Université, CNRS and Région Ile de France. A.-L. R. is greatful to the COST Action CA15209 EURELAX "European Network on NMR Relaxometry", supported by COST (European Cooperation in Science and Technology).

We also acknowledge the transverse R&D program from TOTAL S.A. for full support and resources.

The authors wish to thank Jean-Marc Moron from Total for the kerogen isolates extraction.










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
