# Peer review of "Magnetic expression in kerogen reveals impact on fluid transport"

_Magnetic Resonance, 2022_

## Author Response (AR1)

Benjamin Nicot                                    Pau June 2, 2022
Total Energies
Avenue Larribau
6400 Pau, France

Dear Editor of Magnetic Resonance

We have really appreciated the general positive opinion about the combination of our different techniques, not discussed in conjunction before, for understanding the physical properties of embedded liquids in shale oils rocks.

However, the referee 2 proposes a list of comments about individual facts that seems to be not proven sufficiently. He suggests to significantly extend the manuscript to fully exploit the value of the proposed individual results. In the following, we propose to answer carefully to all these questions sequentially. In all the cases, we remind the referee's comment quoted by a point followed by our response and modification on the manuscript

- As mentioned, "magnetic locking" appears to be a buzzword but it is not substantiated; it should be avoided if it cannot be explained precisely

We do want to avoid the use of buzzwords and therefore we propose a new version of the manuscript without the  word" magnetic locking"

- I wonder whether the abbreviation "EMR" needs to be used, it is not common; I would prefer "EPR" but this could be a matter of taste

We have discussed this and believe that "EMR" is more generic and prefer to keep it.

- The beginning of the introduction may highlight the importance of kerogen for understanding evolution, but it does not seem to constitute the main focus of this work

Yes it is true, at the beginning of the introduction we start with a very broad view of kerogen. We then quickly concentrate on the heart of the subject.

- The presentation on ll. 40ff suggests that the study of water/oil distribution in shales, and understanding of their relaxation properties, has been finalized once and for all – I doubt this is the case. The presented model, like any model, is a suggestion with the result of fitting parameters that may or may not describe the system realistically; alternative models exist, and there is a wide range of shales that may have quite different geometry and composition. The paragraph ends with the quest for describing "transport", yet transport occurs on a wide range of scales in space and time. I would

suggest that the authors present findings on the local scale of a molecular dimension; a macroscopic diffusion coefficient, a transport coefficient or a permeability are not determined and, in my opinion, cannot be inferred from the presented data.

Of course, we have used a definite proposed model of relaxation for interpreting the nuclear magnetic relaxation dispersion (NMRD) as well as 2D correlation spectra T1-T2 of oil and brine embedded in shale rocks to extract some relevant dynamical parameters. However, we have used other experimental techniques that prove the unicity of our modelling procedure.

(i) Electron spin resonance (ESR) quantifies unambiguously the sources of NMR relaxation attributed to Mn2+ in the inorganic part and radicals C° in the organic kerogen part.

(ii) Advanced image processing of zoomed SEM images has confirmed the sponge-like kerogen porosity and lamellar clay structure of mineral matter. In particular, FIB-SEM has evidenced a hierarchy of the pore-size distribution on a large extent. This is of particular importance for the interpretation of the NMRD data, especially at low Larmor frequency, where the longitudinal relaxation rates 1/T1, that is proportional to the specific surface area SP,NMR, become very large.

(iii) For a very large distribution of pore sizes (75-85% of the porosity visible on SEM images), we found unique values of the translational diffusion of oil (Dsurf =2.6 10-7 cm2/s) and water (Dsurf =1.9 10-5cm2/s) at the pore surfaces of kerogen and clays, respectively.

(iv) Last, the biphasic fast exchange condition that ensures the monoexponential relaxation for oil and water, respectively and the magnetic locking occurring in kerogen, show that oil is transferred by diffusion very slowly between large organic pores through solid kerogen. Both effects extend the diffusion range to a larger extent.

From an NMR relaxation point of view, the nuclear magnetic relaxation is due to a modulation of the translational liquid diffusion at proximity of paramagnetic impurities fixed at the surfaces of the pores. Our modelling captures the main point that the probability of reencounters between the mobile probed molecules and the fixed paramagnetic species is drastically enhanced by the low dimensionality of the local geometry. Other geometries of pores could exist in some other shales, but the essential features of the NMR relaxation are fully considered by our modelling, i.e. a liquid that makes numerous back and forward dynamics at proximity of important sources of relaxation.

To clarify these points, we have thus added a whole paragraph after Line 48.

- In the same paragraph (and in the following), it is mentioned again that the low mobility of molecules, and/or the low permeability, in shales is unexpected and unexplained; (i) I tend to say that it is not unexpected and (ii) there are only two references cited in this context, unfortunately both of them not readily available to the reader (Le Bihan not properly cited; Sondergeld a special journal) while there are tons of paper discussing the question of permeability in shale, for obvious commercial reasons, quite a lot of them also aiming at a molecular description – I find this problematic because the authors start out with a hypothesis that does not seem to be obvious to the reader

This is precisely the aim of our paper. How to explain the low permeability of these shale oils? In fact, we have asked other two questions: how to localize fluids either in mineral or organic porosities? How to characterize their individual dynamics? To answer these fundamental questions, we used very different and complementary techniques. All these techniques bring structural and dynamical information on different length and time scales.

We have used also some real imbibition drainage experiment that evidence very fast penetration of oil within the kerogen and very slow water in shales (Nicot B, et al 2015). This transport experiment evidences the very large pore connectivity existing in kerogen compared to a smaller one in clays. Moreover, the real interwinned structure of organic and inorganic parts of the shale exhibits the real impact of the kerogen nanopores in the long-range fluid transport in these materials. This is why the local magnetic locking, described in our paper at the tiny level, could affect the macroscopic transport in such a large interconnected network of pores.

The reference Le Bihan et al has been corrected.

- 2.1. samples: what are the "native fluids"? this is anything but obvious, and rather important. Are these mixtures of water and oil; which aromaticity ratio; has a SARA analysis been made of the oil? These details would be required preferably in this section, or at least in the Supplementary Information

A sentence as been added in paragraph 2.1 to describe the native state and native fluids. The native fluids are the fluids contained in the rock naturally, before we do anything in the lab.  The SARA analysis could be done on the extracted oil, but we did not see the interest in this study so it has not been performed.

- In the same context, please explain the "HCl/HF" demineralization process. If I threw a shale sample into acid, what would I get? What is the recipe? Why are kerogens totally unaffected, or are they – especially their structure needs to be maintained to appreciate the similarity argument from the

HYSCORE experiments. A non-expert would not know this. In general, "sample preparation" is insufficiently described.

The HCL/HF process enables to dissolve all the mineral parts of a rock, leaving the organic parts. A reference (Durand et al. 1980) has been added on line 70.

In previous studies, some co authors have found that the HCl/HF attack does not affect carbon radicals (see Binet et al, Heterogeneous distribution of paramagnetic radicals in insoluble organic matter from the Orgueil and Murchison meteorites, Geochimica et Cosmochimica Acta, 2002).

- 87ff: does this image analysis study a planar surface with depth zero, or does the image represent an average through a depth of xx nm? Could this affect the interpretation?

We assume that the image analysis for the 2D porosity quantification is representative of the sample surface and concerns a planar surface averaged through a negligeable depth, as the penetration of the electrons and the interaction volume is reduced, in the operating conditions choosen for the acquisitions (low acceleration voltage);

- Same paragraph: as I understand it, the automated algorithm measures circumference and pore area – this would be information for determining S/V ratio or a shape factor, although the shape information is dismissed; first computing an equivalent disk diameter and then a fractal dimension from the distribution of these diameters just appears a bit too complicated – how good is the disk representation? (apparently not very good for the clay structure in Figure 2)

We are aware that the disk representation is not perfect, especially for the clay structure, however this simple calculation allows us to have an idea of a pore size distribution with a simple model.

- 2.4: at the end, it should be explained that a separation between brine and oil is only possible after applying particular models, they are not separable per se

In Nicot et al 2015, we have performed NMR Dispersion experiments on shale samples that we saturated with only water, this gave us the water signature. We performed also the same experiment with samples saturated by oil to obtain the oil signature. Therefore, the assignement of signals in figure 8 is unambiguous and does not depend on a model.

- 182: what is the significance of the fractal dimension of the PSD? Is this a continuous function within boundaries, and if it is, what are the lower and

upper limit? I can imagine that a fractal dimension of pore sizes will be reflected in diffusion properties, and possibly also in relaxation properties if the length scale is appropriate – yet the authors do not make use of this information at all in their work

Advanced image processing of zoomed SEM images has confirmed the sponge-like kerogen porosity and lamellar clay structure of mineral matter. In particular, FIB-SEM has evidenced a hierarchy of the pore-size distribution $N(R) \propto R-D_f$ between the following boundaries ($R_{min}$ = 2.5 nm and $R_{max}$= 630 nm), where $D_f$ ~2.3 is the surface fractal dimension which characterizes the self-similarity of the pore geometry between these boundaries.

To clarify these points, we have thus added a paragraph on Line 183

- 193f: no it doesn't. there may well be organic radicals also in the rock - but it is unlikely; however, there could be an underlying quartz defect line at a similar g value which often overlaps with the organic radical line – has this been tested?

E' center can effectively be closed to carbon centered radical even if its g factor is closer to these of free electron g value. E' center generally provide a very narrow that can be detect at a very low microwave power because of it's fast saturation. It can be easy isolated from carnon signal just by changing the detection mode using phase quadrature detection. As $T_{1e}$ is longer for for E' than C°, if E' is present only one line is detected. No E' center have been detected in ours samples.

- 209ff: I can follow the general argument, but since I am not familiar with HYSCORE: what does "low mobility" actually mean, can it be quantified? Can it be translated into a residence time within a given distance, or a characteristic rotation or translation time? The description appears too qualitative

HYSCORE experiments is based on electron spin echo envelop modulation. that require electron nuclear dipolar interaction. if the mobility is to high that can be the case at room temperature the dipolar interaction vanishes and no nuclear frequencies can be detected.

- 228: how can the rather sharp features for the pure kerogen be explained? Why should the Mn ions actually have a rather well-defined distance of 2 nm, why are they not more randomly distributed?

The two measurements presented have been acquired at similar signal to noise ratios and processed with the same alpha parameter for the inversion.

We do assume that Mn are randomly distributed, the total amount of Mn in the given volume leads to an average distance of 2nm. The Mn-Mn distances are calculated from an average distribution (around 2nm), this result is coherent with the peldor measurement and the schematic interpretation of figure 5.

- 235: what is meant by "reinforcing"? Is the effect not merely a change in distances, which leads to an increase of coupling strength due to the distance dependence? I am not sure if I understand how swelling leads to a reduction in these distances – maybe a sketch would help?

You are correct, the word "reinforcing" has been removed and the sentence rephrased

- 252: I suggest to remove the remark about the relation between line shape and age; or supplement it by an accessible reference, if indeed there is such a clear correlation (the given reference is a conference contribution)

We agree that the discussion regarding the age of bitumen does not serve the purpose of the article and removed it. It was based on observations in our labs: the older the rock, the lower the H/C ratio, the less nuclei are interacting with electrons, the lower the Gaussian/Lorentzian ratio.

- 254: I assume that the apparent multiplet (red line) in figure 6b is supposed to be the vanadyl line; have the authors confirmed this by fitting to the expected lineshape, or is this based on the (realistic) assumption that the suspect can only be VO2+? Is it possible to estimate an amount of VO2+ from this spectrum? (see also Figure S10, which may be discussed in more detail)

While vanadium concentration may vary between various geological conditions and redox conditions, vanadium is found in oceanic petroleum systems

- 256ff: what would be the requirement for an antiferromagnetic ordering? Does this indicate a certain (maximum) distance? How frequent is this occurrence, in other worlds – is such an ordering, stemming from two different types of radicals, regularly observed in comparable systems? I am not familiar with this phenomenon, and I feel that it deserves much more explanation because it would represent a major finding of this study. As in several other cases in this paper, unfortunately, this effect is merely mentioned en passant, though the reader may not be able to assess its importance

There is no magnetic ordering, but only antiferromagnetic interactions between neighboring paramagnetic species. It is not possible to deduce a distance between radicals because the antiferromagnetic is transferred by chemical interactions

between radical species. To the best of our knowledge it is the first time that such behavior is observed in amorphous organic matter.

- 282: what is learnt from the $^{29}$Si signature in the native shale sample? Can this be interpreted by average distances of radicals to the solid matrix?

In the native shale sample, the shale contains large amounts of shales. Shales are alumino silicates, and contain a large amount of Silicon. Therefore, interactions with silicon are very likely. In this case it seems difficult to calculate distances, this could be done but has not been done.

In the extracted kerogen, there always exist residues of silicates that have not been dissolved. (See Gourier, Extreme deuterium enrichment of organic radicals in the Orgueil meteorite: Revisiting the interstellar interpretation? Geochimica et Cosmochimica Acta. 2008)

- 319: I do not understand what the authors want to conclude at this point – it seems that the dodecane molecule assumes a particular position with respect to the solid phase; this may or may not be between the mentioned units, thereby suppressing the coupling; even if the molecules "dock" at a particular position, or a preferential position, why can this be considered as "magnetic locking"? (Note: there is some limited body of literature about the respective location of carbon radicals and VO2+, this has been studied with respect to crude oil but possibly also for solids – would be worth going into this and provide citations).

We agree with your comment and modify the manuscript accordingly.

- Figure 8: at this point we see the different dispersions of water and oil in shale; before, it was mentioned as "native fluids" (see comment above), it may be clarified in the experimental session what fluids, and in which composition, are present in the shale; also one might explain why all experiments up to this point do not detect the presence of water. After all, the distribution of water and oil with respect to the surface would be an important parameter

The Figure 8 is a key-point in this manuscript. We evidence first by high resolution NMR of an "as received" shale the presence of separated oil and water peaks. Here, the experimental filled points (red and blue) have been obtained by a Laplace inversion of the longitudinal magnetization decay of an "as received" shale for every Larmor frequency. We observed a net bimodal distribution of T1. The analysis of

the apparition/dispersion of these peaks using different procedures of cleaning the sample have shown that the red points belong to the oil and the blue ones to water. We give in the legends of Fig. 8 the dynamical parameters found for these two fluids with the proposed theory (continuous lines). In inset, we have analyzed the temperature dependence of the longitudinal $T_1$ and transverse $T_1$ relaxation times observed at 20 MHz. Here again, the continuous lines represent the best fits obtained with our theory.

To clarify these points, we have thus added a paragraph on Line 331.

- 354ff: it appears the for the two liquids, fitting to the corresponding equations (in the Supplementary Information) delivers the numbers given in the text. However, I have the impression that at least some of them were determined independently; one reason certainly is that the parameters appear as products in eq. 5 and 6. Text following eq. 5 clarifies that the specific surface area for clay is perhaps taken from a reference (or is it?). Following eq. 6 the specific area and the radius R are given values, while in l. 368f they are suggested as results – this is inacceptable and need to be clarified. Also, the diffusion coefficient is found under (iv) as a particular value but there is no mentioning how it is derived, neither in the main text or in the SI.

All the parameters were found from the best fits of our NMRD data. The impression of the referee is due to a clumsiness of style from our part. Of course some parameters are well known such as the molecular size $\delta$, the different densities $\rho$,... The specific surface area is found from the fits and correspond with the results found in the literature (case of clays). Basically, the typical form of the NMRD profiles allows finding the two correlation times: the surface translational correlation time $\tau_m$ of the liquid and $\tau_s$ is the time of residence of the liquid molecule at the pore surface (Korb et al , 2009). The translational diffusion coefficients are given by the Einstein relations: $D_{surf} = \delta^2/4\tau_m$.

To clarify these points, we have thus added a paragraph on Line 355.

- As mentioned earlier, even if we believe that this D is the correct macroscopic value, since it is derived from microscopic processes on a nm-scale, I do not necessarily agree that this is a particularly low value, given the possibly high tortuosity of kerogen

*See Answer above.* This value of the surface diffusion of oil is representative to the whole pore size distribution. As the pore surfaces are chemically equivalent for the

whole pore size distribution that is very large, there is no reason why there will be differences in the translational diffusion at pore surfaces

A phrase has been added on Line 369 to clarify this point.

- It should indeed be clarified that these two models deliver correlation times, and the fact that the function either fits or not; the value of the models then is in the interpretation of the fitted correlation times

The surface translational correlation time $\tau_m$ of the liquid and the time of residence $\tau_s$ of the liquid molecule at the pore surfaces (Korb et al , 2009) are intrinsically considered in the Eqs. (5 and 6) from the basic features of the NMR relaxation model.

A phrase has been added on Line 379 to clarify this point.

- At some stage, the authors should at least mention that PFG NMR allows the measurement of self-diffusion coefficients; if that were not possible for shales for a particular reason, might be rather interesting for the reader to learn

Basically, the PFG-NMR allows the measurement of self-diffusion in bulk (not at pore surface) when the translational diffusion is the unique process responsible of the dephasing of spins. In other words, this supposes that there is no influence of the relaxation. This is not the case in shales due to the large contribution of the paramagnetic species.

A phrase has been added on Line 380.

- 382: the data are not "dispersed", they are just scattered! Dispersion is the systematic variation of R1 with frequency, but at low frequency R1 probably turns into a constant. In fact, one might comment on why the data for oil are so scattered, and less so for water

It is not only a question of wording. According to Eq. 6, at low frequency the asymptotic theoretical expression for $\frac{1}{T_{1(\omega_I)}} \propto \sqrt{\tau_m \tau_S}$ is a constant in frequency but dependent on $\tau_S$ which is highly dependent on temperature. For an activated process, one has $\tau_S \propto exp(E_S/RT)$ where $E_s$ is the activation energy of the surface interaction. The scattered constant values (in frequency) values $0.4~\mu s < \tau_S < 2.1~\mu s$ are thus consistent with the simulations of Lee et al

(Lee, et al 2016) who consider a wide distribution of residence times for oil in kerogen nanostructure that inhibits the activated desorption of oil. On the other hand, at high frequency, $\frac{1}{T_{1(\omega_I)}} \propto \sqrt{\frac{\tau_m}{\omega_I}}$ is independent of $\tau_S$ with absence of scattering.

A paragraph has been added on Line 385 to clarify this point.

- conclusions around l. 390: as mentioned above, I beg to differ; this is a suggestion, not a proof; the connection to permeability is not explained either

We agree and removed the direct link to permeability.

- the same holds for the actual "Discussion/Conclusion"
- figure 10 would be better in place at an earlier stage of the paper
- 430f: this outlook is pure speculation and unnecessary

While we agree that this is just an outlook; interactions between paramagnetic species in fossil organic matter and associated inorganic phases can have a great interest in various domains such as the ones cited.

- reference list: in addition to the individual literature comments made above, there is certainly a bunch of relevant literature of people having accumulated knowledge of fluid dynamics in porous rocks and shales over the decades, some of the deserve to be cited in order to put this research into better perspective

Supplementary Material:

- Please improve quality of figures S1 (larger, clearer scale bars) and S2 (text inside figure)

- 72: would the value of k be the same for any kind of interface and any temperature? How is it estimated?

The dipolar cross-relaxation rate from the confined liquid to the solid proton species has been chosen as k∽1s$^{-1}$ for a good fit. This value is consistent with an asymptotic limit at low frequency which tends to constant (independent of the frequency) at low frequency.  k is thus limited by the transfer of dipolar energy (spin diffusion) within the bound solid proton. The

observed plateau below a frequency $\omega_c/2\pi \sim 25$ kHz is thus characteristic of the rigid-lattice limit of the solid proton hydrates $R_{sol}$.

A phrase has been added on Line 73 to clarify this point.

- eq. 2: if F<<1, then the term 1/F would dominate in each occurrence in this equation – is this correct? What is the consequence?

F<<1 is the ratio of the solid-proton population to the liquid proton at equilibrium (Lester et all 1991), It is thus necessary very small compared to one.

- In general: eq. 1 is a representation of a two-component situation with exchange; on the other hand, this approach is seldom used for liquid relaxation in porous solids – why is this the case? Are most people doing it wrong? Which approximations can be made to allow the "simple" approach (with Brownstein/Tarr averaging)? I assume that the two alternative approaches eq. 5 and 6 serve to compute the "liquid" contribution according to eq. 3 which also appears in eq. 2. If the experimentally observed results are fitted with the full equation 2, then one would need to know the "solid" contribution of R1, or rather: its frequency dependence. However, I don't think this has been done, so the R1solid term may be neglected – but if the authors cite eq. 2, they need to explain all relevant contribution, including R1solid – is this really the value that is mention as the low-frequency plateau in line 126? Then, what exactly are the protons in the solid phase, especially in the mineral phase?

 Yes, Eq. 1 is a representation of two-component situation with exchange. Even if this approach is seldom used for liquid relaxation in porous solids. This is not the general case, see for instance the case of relaxation in cement-based materials *(see Barberon, F; Korb, JP, et al Phys rev. Lett. 90 116103 (2003)*). If we neglect the exchange, $k \rightarrow 0$, we find the simple approach of Brownstein and Tarr. However, in that case, it is impossible to find a plateau of $1/T_1 \sim R_{1,sol}$ at very low frequency and in that case Eqs. 5 or 6 diverge at low frequency. We have thus fitted our NMRD data with the full Eq. 2 as well as Eqs. 5, 6. The values chosen for $R_{1,sol}$ should correspond to $1/T_1(\omega_I < \omega_c \sim 2\pi \; 25\text{kHz}) \sim 500\text{-}600 \; \text{s}^{-1}$. These values are typical to the numerous hydrates present in the shale samples.

A paragraph has been added on Line 73to clarify this point.

- Figure S5 is difficult to read. Usually, T1/T2 is a diagonal line parallel to the main diagonal. There is such a line but it is not labelled. On the other hand,

there are three averages <T1/T2> (how have these averages been computed? For which field strength?). I do not understand which part of the 2d spectrum these three averages relate to; I do not understand how the experimental data are similar to the theoretical prediction (dashed and full lines); colors should be the same for each field strength.

$T_1/T_2 \sim 2$ is only a diagonal line for a viscous liquid for a weak confinement close to the pore surface like in a rock sample of large pores. However, in presence of a highly confinement with dynamical motion in a 1D or 2D pore system, the ratio $T_1/T_2$ is drastically enhanced. According to Eqs. 5, 6, one has $T_{1(\omega_I)} \propto \sqrt{\frac{\omega_I}{\tau_m}} = Cte$ whose value depends drastically on the Larmor frequency while $T_2 \propto 1/\sqrt{\tau_m \tau_S}$, has almost no frequency dependence. This is what we observe in Fig. S5. The ratio $T_1/T_2 = 17$ and 74 at 2 and 23 MHz for oil while it is about 5.5 for water. The oil peaks in the 2D $T_1$-$T_2$ spectrum moves substantially more than the water peak with increasing frequency. This is consistent with a dynamic of oil on a quasi 1D pore in kerogen compared with a 2D dynamics of water in clay.

To clarify this point a paragraph has been added on Line 142

We thank again the Referee 2 for having reading our manuscript so carefully and hope that